# Fabrication of PA-PEI-MOF303(Al) by Stepwise Impregnation Layer-by-Layer Growth for Highly Efficient Removal of Ammonia

**DOI:** 10.3390/nano13040727

**Published:** 2023-02-14

**Authors:** Liang Lan, Xuanlin Yang, Kai Kang, Hua Song, Yucong Xie, Shuyuan Zhou, Yun Liang, Shupei Bai

**Affiliations:** 1School of Light Industry and Engineering, South China University of Technology, Guangzhou 510640, China; 2State Key Laboratory of NBC Protection for Civilian, Beijing 102205, China

**Keywords:** NH_3_ capture, metal-organic frameworks, polyacrylate macroporous polyester, stepwise impregnation layer-by-layer growth, hierarchical pore composite

## Abstract

NH_3_ is a typical alkaline gaseous pollutant widely derived from industrial production and poses great risks to humans and other biota. Metal-organic frameworks (MOFs) have excellent adsorption capacities relative to materials traditionally used to adsorb NH_3_. However, in practice, applications of MOFs as adsorbents are restricted because of its powder form. We prepared a polyamide (PA) macroporous polyester substrate using an emulsion template method and modified the surface with polyethylenimine (PEI) to improve the MOF growth efficiency on the substrate. The difficulty of loading the MOF because of the fast nucleation rate inside the PA macroporous polyester substrate was solved using a stepwise impregnation layer-by-layer (LBL) growth method, and a PA-PEI-MOF303(Al) hierarchical pore composite that very efficiently adsorbed NH_3_ was successfully prepared. The PA-PEI-MOF303(Al) adsorption capacity for NH_3_ was 16.07 mmol·g^−1^ at 298 K and 100 kPa, and the PA-PEI-MOF303(Al) could be regenerated repeatedly under vacuum at 423 K. The NH_3_ adsorption mechanism was investigated by in situ Fourier transform infrared spectroscopy and by performing two-dimensional correlation analysis. Unlike for the MOF303(Al) powder, the formation of multi-site hydrogen bonds between Al–O–Al/C–OH, N–H, –OH, C=O, and NH_3_ in PA-PEI-MOF303(Al) was found to be an important reason for efficient NH_3_ adsorption. This study will provide a reference for the preparation of other MOF-polymer composites.

## 1. Introduction

NH_3_ is a typical gaseous alkaline pollutant that is mainly emitted during agriculture, particularly animal husbandry [1,2,3,4]. NH_3_ is also widely used during industrial processes, including fertilizer and food production processes, and in refrigeration equipment [5,6,7]. NH_3_ is considered to have a high hazard index by the North Atlantic Treaty Organization (NATO) because of its toxicity and the risks associated with industrial mass production and storage of NH_3_ [8]. Humans experience sensory discomfort at NH_3_ concentrations of a few parts per million, and NH_3_ concentrations >35 ppm pose serious risks to humans and other animals [9,10]. Moreover, NH_3_ released into the air can combine with NO_x_ or SO_x_ to form ammonium salts, leading to the formation of PM_2.5_ [11,12]. It is often difficult to efficiently remove NH_3_ using conventional absorption methods. Therefore, it is important to develop new porous materials for removing NH_3_ to protect human health and the environment.

As the traditional porous materials, activated carbon and zeolites are widely used to adsorb NH_3_ [13,14]. NH_3_ adsorption by such porous materials is mainly dependent on the pore structure and the presence of any active constituent impregnated into the porous substrate, but the adsorption capacities of most such materials are low [15,16,17]. Metal-organic frameworks (MOFs) have very high specific surface areas, adjustable porous channel structures, and skeleton modifiability that can provide ideal spaces for accommodating various gases. MOFs have attracted much interest as new adsorbent materials [18,19,20]. Generally, NH_3_ and H_2_O can cause the structures of MOFs to collapse [21,22,23]. But as Jasuja et al. reported, the stability of MOFs can be significantly enhanced by placing nonpolar shielding groups (e.g., methyl) on the BDC linker [24]. Wang et al. constructed a structurally stable MOF303(Al) by a multisite ligand screening strategy. Its NH_3_ adsorption capacity was up to 19.7 mmol·g^−1^ and the NH_3_ capture was completely reversible [25]. MOFs are usually present in powder form. In applications, MOFs need to be compounded with other materials to allow fibers, films, membranes, or particles to be prepared [26,27,28]. Therefore, there are challenges when preparing high-performance MOF composites without losing the original properties of MOFs.

MOFs and porous polymer composite are an emerging material and have been extensively studied in recent years [29,30]. Porous polymers have the advantages of modifiable pore structures and easy processing and molding, and are widely used in various fields. The preparation of polyacrylate (PA) polymers with interpenetrating macroporous structures by emulsion template polymerization has attracted great interest as a family of porous polymer materials [31,32]. It has the advantages of low density, good mechanical strength, controllable pore structure, and a simple preparation process, which have the potential for practical engineering applications [33,34]. Furthermore, the diverse monomers of porous polymers that can be used can provide a wide range of functional groups. Compared with other substrates, PA macroporous polymers have better compatibility with MOFs [35,36].

Hybridization of MOFs with organic polymers is a challenge in the preparation of MOFs/polymer composite [37,38,39]. MOFs and porous polymer composite are synthesized using strategies involving a mixing method and in-situ growth [40]. A mixing method is mainly used to produce MOFs/polymer composite from pre-synthesized MOFs that become adhered to or encapsulated in the polymer during a molding process. This gives a convenient and fast synthesis method, but has some disadvantages, including the MOFs being poorly dispersed in the emulsion, the MOFs pores readily becoming blocked, and the physical strengths of the MOFs nodes in the composite readily becoming degraded [41,42]. In-situ growth means heterogeneous growth of MOFs on a substrate. This tends to preserve the physical and chemical properties of the MOFs and the polymer [43,44]. However, this method often requires the substrate to be pre-modified by functional group grafting or atomic layer deposition to improve the compatibility of the substrate and MOFs, and the MOFs have an uneven growth or low loading [45,46].

In this study, a PA skeleton structure with an interpenetrating macroporous structure was synthesized by emulsion suspension polymerization. Surface functionalization with polyethylenimine (PEI) improved the compatibility between the PA substrate and the MOF. A PA-PEI-MOF303(Al) composite with a multi-level pore structure was successfully constructed by introducing MOF303(Al) into the PA skeleton using a stepwise impregnation LBL method [47], which prevented uneven and low loadings that are often achieved using a one-step in-situ growth method. The physical and chemical properties of the PA-PEI-MOF303(Al) composite were analyzed. NH_3_ adsorption was evaluated using a breakthrough method and PA-PEI-MOF303(Al) exhibits an excellent NH_3_ adsorption performance. Finally, the NH_3_ adsorption mechanism was systematically investigated. PA-PEI-MOF303(Al) composite material prepared using stepwise impregnation LBL method not only solved the difficulty of the MOFs molding problem, but showed excellent adsorption capacity of NH_3_.

## 2. Materials and Methods

### 2.1. Materials

Tert-butyl methacrylate (99%), glycidyl methacrylate (97%), trimethylolpropane triacrylate (85%), BPO (toluoyl peroxide), methyl alcohol (99%), cetyl alcohol, polyvinyl alcohol 1788 (PVA), ammonium persulfate (APS), N,N-dimethylaniline, tetramethylethylenediamine, polyethyleneimine (M_W_. 1800.99%), aluminum chloride hexahydrate (99%), 3, 5-Pyrazoledicarboxylic acid monohydrate (97%), and sodium hydroxide (97%) were purchased from Shanghai Maclean’s Reagent Co., Ltd., China. Poly(ethylene glycol)-block-poly(propylene glycol)-block-poly(ethylene glycol) (PEG-PPG-PEG, Mw = 5800) was purchased from Sigma Aldrich (Shanghai) Trading Co., Ltd, China. Toluene (AR) was purchased from Tianjin Jindong Tianzheng Fine Chemical Reagent Factory. All chemical reagent materials were used directly without pretreatment. NH_3_ (1%) mixture gas was purchased from Beijing Haipu Gas Co., Ltd., China.

### 2.2. Synthesis Method

#### 2.2.1. Synthesis PA Substrate and the Modified Material

The PA porous polyester substrate material was prepared using the procedure described in the literature [48]. The type and amount of reagents were adjusted for specific experiments. The porous polyester substrate material was prepared by an emulsion suspension polymerization method. Polyvinyl alcohol (2.0 g) and ammonium persulfate (0.48 g) were added into a three-neck flask containing 300 mL of deionized water and dissolved with stirring to obtain the third-phase solution. Tert-butyl methacrylate (1.4 g), glycidyl methacrylate (4.6 g), trimethylolpropane triacrylate (4 g), BPO (0.2 g), cetyl alcohol (0.2 g), Poly(ethylene glycol)-block-poly(propylene glycol)-block-poly(ethylene glycol) (1.2 g), and toluene (7.8 g) were mixed and dissolved by ultrasonication to form an oil phase. A total of 30 mL of deionized water was added dropwise to the oil phase in the ice bath within 5 min and mixed with a high-speed disperser at 3000 rpm. The oil phase was then stirred at 6000 rpm for 10 min (1 mL of N, N-dimethylaniline was added in the last 1 min) to obtain an emulsion. The prepared emulsion was added to a three-neck flask containing the third phase, followed by the rapid addition of 0.5 mL of tetramethylethylenediamine. Polymerization was carried out at 343 K for 30 min at 250 rpm and N_2_ was purged into the three-neck flask to protect the polymerization. After the reaction, the PA substrate of 0.6–1 mm particle size was screened with a sieve, washed twice with water, soaked in ethanol for 12 h, and finally dried in an oven at 343 K until constant weight.

A certain amount of PA substrate was placed in a round-bottom flask with a pressure-equalizing funnel with double PTFE stopcocks attached to the upper end. The PA substrate was treated under vacuum at 393 K for 2 h. PEI and methanol were dissolved well by sonication at a mass ratio of 4:1 and then transferred into the round-bottom flask under vacuum. After complete impregnation, the reaction was carried out in nitrogen atmosphere at 403 K for 2 h. The modified material (PA-PEI) was obtained by being washed with water to neutral, soaked in ethanol for 12 h, and dried in an oven at 343 K to constant weight. The reaction process of the synthesis of PA and PA-PEI is shown in Figure 1.

#### 2.2.2. Synthesis of PA-PEI-MOF303(Al)

PA-PEI was placed in a round-bottom flask with a pressure-equalizing funnel with double PTFE stopcocks attached to the upper end and treated under vacuum at 393 K for 2 h. A 1.08 M metal salt solution was prepared by dissolving 5.2 g aluminum chloride hexahydrate in 20 mL deionized water. Then the metal salt solution was added to the round-bottom flask under vacuum and sonicated for 5 min. After aging for 24 h, the salt solution was filtered out. The PA-PEI-Al modified substrates were obtained by being dried in an oven at 343 K until constant weight. Similarly, the PA-PEI-Al material was vacuum-treated at 393 K for 2 h. An amount of 1.1 g of 3, 5-pyrazole dicarboxylic acid monohydrate was dissolved in 30 mL of deionized water, then 3 mL of sodium hydroxide solution (2.57 M) was added and sonicated until the ligand was well dissolved to prepare 0.19 M ligand solution. The ligand solution was then added to a round-bottom flask containing a PA-PEI-Al substrate under vacuum and sonicated for 5 min. Subsequently, the matrix and ligand solution were all transferred to a Teflon-lined autoclave reaction kettle and reacted at 373 K for 24 h. After the system was cooled to room temperature, the composite was washed with methanol using a Soxhlet extraction device for 24 h and dried under vacuum at 343 K for 24 h. As shown in Figure 2, PEI-PA-MOF303(Al) composite was prepared through five times of impregnation and regrowth processes. 

### 2.3. Characterization Methods

The X-ray diffractometer (XRD) characterization was performed on a 3 kW Ultima IV (Rigaku, Japan) with Cu–Kα radiation (λ = 1.5418 Å) over an angular range of 2θ = 5–90° at a scan rate of 5°·min^−1^. The Brunauer-EmmettTeller (BET) specific surface area was obtained on an Autosorb IQ surface area analyzer (Quantachrome, USA) under an N_2_ atmosphere at 77 K. The morphology was photographed by SEM (ZEISS GeminiSEM 300, Germany) with an accelerating voltage of 0.02–30 kV, continuously adjustable in 10 V steps, and a probe beam current of 3 pA–20 nA. Thermogravimetric analysis (TGA) was performed (Rigaku TG/DTA8122, Japan) under a nitrogen atmosphere with a heating rate of 5 K·min^−1^. Fourier transform infrared spectroscopy (FTIR) was recorded in a range of 400–4000 cm^−1^ (PerkinElmer Frontier, USA).

### 2.4. NH_3_ Adsorption Performance Test

#### 2.4.1. Adsorption Isotherms

Adsorption isotherms are empirical relationships that explain the extent and mechanism of adsorption. They are mainly used to describe the relationship between the adsorption capacity and the reaction pressure or adsorbent concentration at a specific temperature. In this paper, the adsorption isotherms of ammonia gas on PA-PEI-MOF303(Al) composite were tested by the volumetric method, and the adsorption isotherm data were fitted with Langmuir and Freundlich isotherm models to determine the action occurring within the adsorption system and to elaborate the adsorption mechanism of ammonia gas on the composite [49,50].

The non-linear form of the Langmuir isotherm model equation is given below:(1)$$q_{e}=\frac{Q_{m}K_{L}C_{e}}{1+K_{L}C_{e}},$$
where $q_{e}$ (mmol·g^−1^) is the equilibrium adsorption amount, $Q_{m}$ (mmol·g^−1^) is the monolayer limiting adsorption capacity, and $K_{L}$ (L·mmol^−1^) is the Langmuir adsorption constant.

The non-linear form of the Freundlich isotherm model equation is given below: (2)$$q_{e}=k_{f}Ce^{n_{f}},$$
where $K_{f}$ is the Freundlich adsorption constant and $n_{f}$ indicates the heterogeneity of the data distribution of the active centers, which is a measure of the adsorption intensity.

#### 2.4.2. Adsorption Thermodynamics

The adsorption mechanism can be explained by thermodynamic parameters, such as Gibbs free energy (ΔG°), enthalpy change (ΔH°), and entropy change (ΔS°). In this paper, ΔG°, ΔH°, and ΔS° are calculated based on the adsorption curves obtained by fitting the Freundlich model (*R*^2^ > 0.99) [51,52]. The calculations were performed as follows:(3)$$\Delta G{}^{\circ}=-RT\ln k_{e}{}^{\circ},$$
(4)ke°=kf1nf,
(5)ΔG°=ΔH°−TΔS°,

The Van’t Hoff equation is obtained by putting Equation (3) in the position within Equation (5):(6)$$\ln k_{e}{}^{\circ}=\frac{\Delta S{}^{\circ}}{R}-\frac{\Delta H{}^{\circ}}{R}\frac{1}{T},$$
where R (8.314 J·mol^−1^·K^−1^) is the universal ideal gas constant and T(K) is the absolute temperature. ΔG°(KJ·mol^−1^) is calculated directly from Equation (3). ΔH°(KJ·mol^−1^) and ΔS°(J·mol^−1^·K^−1^) can be calculated by the slope and intercept of $\ln k_{e}{}^{\circ}$ to 1/T.

The isosteric heat of adsorption for the composite was calculated by applying the Clausius–Clapeyron (CC) equation on isotherm data at two temperatures [53,54]:(7)$$q_{iso}=-\Delta_{ads}H_{diff}=\frac{RT_{1}T_{2}}{T_{2}-T_{1}}\ln\left(\frac{P_{2}}{P_{1}}\right),$$
where R is the gas constant, $T_{1}$ and $T_{2}$ are the adsorption temperatures, and $P_{1}$ and $P_{2}$ are the respective absolute pressures at a given loading.

#### 2.4.3. Dynamic Adsorption Performance Measurements

The breakthrough experiments were used to evaluate NH_3_ dynamic adsorption performance. The self-constructed NH_3_ dynamic adsorption evaluation system is shown in Figure 3. Dynamic adsorption performances of the prepared materials were evaluated in a vertical fixed bed reactor with a quartz tube of 8 mm inner diameter. The materials with height of 8 cm were loaded into the quartz tube and the experiment temperature was 298 K. The reaction gas of 500 ppm NH_3_ was generated by mixing cylinder gas (1% NH_3_, 99% N_2_) with the purified compressed air (relative humidity < 5%) at a total gas flow rate of 0.1 L·min^−1^. An infrared gas detector (Beijing BAIF-Maihak Analytical Instruments Co., Ltd., QGS-08C, Beijing, China) was used for online analysis. The materials were heated at 423 K under vacuum for 2 h before the breakthrough experiments. For the cyclic adsorption-desorption experiment, the regenerated materials were obtained by heating the saturated adsorbent materials at 423 K for 2 h under vacuum conditions.

### 2.5. In Situ FTIR Spectroscopy Study and Two-Dimensional Correlation Spectroscopic Analysis

In situ FTIR spectroscopy was used to study the adsorption and heating-desorption processes of ammonia on PA-PEI-MOF303(Al) composite. Before testing, the material was activated at 423 K under vacuum for 2 h and cooled to room temperature by purging for 1 h under argon atmosphere. The adsorption process was then continued for 1 h with 0.1% ammonia gas introduced at 30 mL·min^−1^. Subsequently, gas introduction was stopped and the PA-PEI-MOF303(Al) composite was heated at a heating rate of 5 K·min^−1^.

To understand the adsorption behavior of ammonia on PA-PEI-MOF303(Al) composites, 2DCOS-FTIR analysis was performed with the adsorption time and temperature variation as external perturbations [55]. The 2DCOS-FTIR analysis was produced by using the 2DShige software released by Kwansei-Gakuin University, Japan [56]. More detailed information of the algorithm adopted in the software has been described by Noda and Ozaki [57].

## 3. Results

### 3.1. Synthesis of PA and PA-PEI

An optical picture and internal surface images of the porous PA polymer substrate prepared by emulsion polymerization are shown in Figure 4a,b. The porous PA polymer substrate is almost spherical and has spherical pores with 2–20 μm interpenetrating macropores [25,58]. Obviously, the pore structure of PA did not change (Figure 4c), but the hydrophilicity changed after PEI modification (Figure 4d,e). The increase in hydrophilicity caused by modifying the PA with PEI favored subsequent mass transfer of MOFs from a solution that entered the pore channels. The FTIR spectra of the PA substrate before and after modification are shown in Figure 4f. In the spectrum of PA, the typical C=O stretching vibration peak of polyacrylate at 1732 cm^−1^ is present [59]. The three characteristic peaks assigned to tert-butyl methacrylate, –CH stretching vibration, and the –CH_2_ shear vibration appear at 1734 cm^−1^, 2975 cm^−1^, and 1467 cm^−1^, respectively [60,61]. While peaks at 905 and 804 cm^−1^ were assigned to vibrations of the skeletons of ternary ring ethers in glycidyl methacrylate [62]. Obviously, the cyclic ether skeleton vibration peak of the PA-PEI substrate was significantly weaker than that of the PA substrate, indicating that the cyclic ether structure was disrupted in the PA-PEI substrate. There is a peak at 1566 cm^−1^ in the PA-PEI substrate, which was designated as N–H bending vibration, indicating that the PEI had been successfully grafted onto the PA substrate [63].

### 3.2. Synthesis of PA-PEI-MOF303(Al)

The FTIR spectra are shown in Figure 5a. A peak in the MOF303(Al) spectrum at 3394 cm^−1^ was assigned to N–H stretching vibrations [64], peaks at 3146 and 3009 cm^−1^ were assigned to aromatic C–H stretching vibrations, a peak at 1529 cm^−1^ was assigned to C=N stretching vibrations, a peak at 1000 cm^−1^ was assigned to N–NH vibrations [65,66], and peaks at 1393 cm^−1^ and 1607 cm^−1^ were assigned to coordination carboxyl groups of –COO–Al [67]. The MOF303(Al) and PA-PEI-MOF303(Al) FTIR spectra each contained a peak at 1601 cm^−1^ that was not present in the PA-PEI spectrum. This was the characteristic peak of COO– in MOF303(Al), indicating that the MOF was successfully grafted onto the PA-PEI. The X-ray photoelectron spectra for MOF303(Al) and the PA-PEI-MOF303(Al) composite are shown in Figure 5b. The spectra indicated that the Al 2p binding energy was 74.91 eV in the MOF303(Al) but 74.41 eV in the composite, indicating that the electron cloud density around Al was different in the MOF303(Al) and composite. This reflected Al–O clusters forming solid connections with amine groups when Al salts became enriched on the PA-PEI substrate surfaces during the composite preparation process. The XRD patterns are shown in Figure 5c. Broad peaks indicated that the PA and PA-PEI substrates were amorphous but the PA-PEI-MOF303(Al) composite gave sharp peaks and 011 and 022 peaks characteristic of MOF303(Al), indicating that MOF303(Al) crystals had been loaded on PA-PEI [68].

In the reported protocol, the MOF303(Al) crystal is in a cubic shape [69]. However, the morphology of MOF303(Al) is cuboid in the PA-PEI-MOF303(Al) composite. We constructed PA and PA-PEI substrate models and investigated molecular collisions, adsorption and molecular transfer at the interface based on first-principles molecular dynamics methods using CP2K/Quickstep software [70]. The growth of MOF303(Al) on PA and PA-PEI substrate surfaces was simulated (Figure 6a). Growth of MOF303(Al) decreased more slowly on the PA-PEI substrate than on the non-surface-modified PA substrate over time, meaning heterogeneous growth of MOF303(Al) would have occurred more readily on the PA-PEI surfaces than on the PA surfaces. The gyration radius indicated growth of MOF303(Al) at the nucleation sites in different directions (Figure 6b). The gyration radius was higher in the x-direction than the y- and z-directions, i.e., the MOF303(Al) tended to grow on the composite somewhat linearly into rod-like structures during in-situ growth, consistent with the cuboid-like crystal clusters found in the PA-PEI-MOF303(Al) composite (Figure 7c,d).

### 3.3. Characterization of PA-PEI-MOF303(Al)

#### 3.3.1. Microstructure

The PA-PEI-MOF303(Al) composite surface and interior morphologies are clearly shown in Figure 7. The loading rate of MOF303(Al) is 51.15 wt% (Appendix A). Unlike other spherical MOF/polymer composites, the PA-PEI-MOF303(Al) composite had a good degree of sphericity (Figure 7a) and a low density of 0.35 g·cm^−3^ (Appendix A), and could be mass-produced [71]. The particle diameters of a PA-PEI-MOF303(Al) ball were nearly 0.6–1 mm. MOF303(Al) formed uniform and dense 0.5–3 μm-long cuboid-like crystal clusters on the PA-PEI substrate surfaces (Figure 7b) and in the internal pore channels (Figure 7c,d), and formed a multi-level pore structure that allowed gas diffusion and mass transfer to the surface and within the composite structure. The EDS maps shown in Figure 7e–g indicated that Al and N were uniformly distributed through the cross-section of the material.

#### 3.3.2. BET Analysis

An N_2_ adsorption-desorption test was used to analyze the BET specific surface area (S_BET_) and pore structures of the PA-PEI substrate and the PA-PEI-MOF303(Al) composite (Figure 8). The physical characteristics of the substrate and composite are shown in Table 1. As shown in Figure 8a, the N_2_ adsorption-desorption isotherm of PA-PEI at 77 K was a typical type IV isotherm. An obvious hysteresis loop during the adsorption and desorption process was found, indicating that the PA-PEI contained mesopores. The BET specific surface area of the PA-PEI substrate was 24.2 m^2^·g^−1^, and only a few micropores were found (Figure 8c). The total pore volume was 0.22 cm^3^·g^−1^. The N_2_ adsorption-desorption isotherm for the PA-PEI-MOF303(Al) composite at 77 K was also determined (Figure 8a). The adsorption capacity increased rapidly at a low p/p_0_ value, which is typical of a type I adsorption isotherm. An obvious hysteresis loop started to appear at p/p_0_ ≈ 0.42, which is typical of a type IV adsorption isotherm for a mesoporous material. The BET specific surface area of the PA-PEI-MOF303(Al) composite was 302.8 m^2^·g^−1^. The micropores mainly had diameters of 0.45–0.80 nm, and the micropore volume was 0.10 cm^3^·g^−1^ (Figure 8b). The total pore volume was 0.279 cm^3^·g^−1^, meaning micropores accounted for 36.6% of the total pore volume. The mean pore diameter of MOF303(Al) was 0.59 nm, the total pore volume was 0.279 cm^3^·g^−1^, and the BET specific surface area was 1292 m^2^·g^−1^ [25]. The BET analysis results indicate that the PA-PEI-MOF303(Al) composite retained the microporous structure of MOF303(Al) and is suitable for adsorbing NH_3_ (molecular diameter of 0.36–0.38 nm) within the micropores.

#### 3.3.3. Thermal Stability

The stability of a material is an important indicator of the ability of the material to be used in any particular application. The thermal stabilities of the PA, PA-PEI, MOF303(Al), and PA-PEI-MOF303(Al) were determined by thermogravimetry, and the results are shown in Figure 9. The PA substrate lost mass slowly at 473 K, which corresponds to the thermal decomposition of the residual components not involved in the polymerization, while the second thermal weight loss after 673 K belongs to the degradation of the polymer substrate. There was a mass loss of about 4% in the PA-PEI substrate from 298 to 373 K. This is because PA-PEI had adsorbed gases such as carbon dioxide and moisture from the air. Subsequently, the PA-PEI substrate decomposed at 473 K [48]. At 293 to 373 K, MOF303(Al) had a small mass loss, which was attributed to the loss of residual solvent in the pores [72]. The PA-PEI-MOF303(Al) composite had a slow mass loss up to 673 K, which was the same as the mass loss of the substrate and MOF303(Al).

### 3.4. NH_3_ Adsorption Tests

#### 3.4.1. Adsorption Isotherms

The adsorption-desorption isotherms of the PA-PEI-MOF303(Al) composite at 298, 313, and 353 K were determined. As shown in Figure 10a, the adsorption-desorption isotherms at all three temperatures were of type I, and the amount of NH_3_ adsorbed increased rapidly in the low-pressure region, indicating that micropores with diameters of 0.45–0.80 nm in the composite adsorbed NH_3_ well. The saturated vapor pressure of NH_3_ increases as the temperature increases from 298 to 353 K. At a constant absolute pressure, the adsorption capacity decreased as the temperature increased (Figure 10a). NH_3_ capture performances on various porous materials were summarized in Table 2. It is worth noting that the NH_3_ adsorption capacity of the PA-PEI-MOF303(Al) composite was a little lower than MOF303(Al), which is due to the influence of the PA-PEI substrate. NH_3_ capture capacity of the PA-PEI-MOF303(Al) composite is about twice that of conventional materials such as activated carbon and zeolite. It is comparable to the NH_3_ adsorption capacity of materials such as COF and MOFs. This indicated that the PA-PEI-MOF303(Al) composite had excellent NH_3_ adsorption properties. The unclosed adsorption-desorption curve indicated that desorption of NH_3_ from the composite was incomplete and that some of the NH_3_ interacted strongly with the composite at each test temperature. This indicated that NH_3_ adsorption by the PA-PEI-MOF303(Al) composite was mainly physical adsorption but that some chemical adsorption occurred.

We fitted Langmuir and Freundlich adsorption isotherms to the data, as shown in Figure 10b, and the fitting results are shown in Table 3. The nonlinear fits were good for both the Langmuir and Freundlich equations, the correlation coefficients *R*^2^ being >0.96. The maximum adsorption equilibrium capacity $Q_{m}$ determined using the Langmuir model was 16.07 mmol·g^−1^ at 298 K, and the adsorption equilibrium capacity at 313 and 353 K were 10.53·g^−1^ and 7.28 mmol·g^−1^, respectively. The adsorption equilibrium capacity decreased as the temperature increased. The Langmuir constants $K_{L}$ were all between 0 and 1, and the Freundlich equation gave 0.1 < $n_{f}$ < 0.5, indicating that adsorption of NH_3_ was favorable at <353 K [50]. The Langmuir constant $K_{L}$ increased and then decreased as the temperature increased. In a certain temperature range, increasing the temperature could generate new adsorption sites and increase chemical interactions between NH_3_ and the PA-PEI-MOF303(Al) composite, resulting first in the increase and then in the decrease of the NH_3_ adsorption rate [78].

#### 3.4.2. Adsorption Thermodynamics

Experiments were performed at 298, 313, and 353 K to determine the thermodynamic parameters of the PA-PEI-MOF303(Al) composite. A better nonlinear fit was found for the Freundlich equation than for the Langmuir equation, and ΔG was calculated using Equations (3) and (4), and the parameters were determined by fitting the Freundlich equation to the data. ΔH and ΔS were calculated from the slope and intercept of a plot of $\ln k_{e}$ against 1/T. The data are shown in Figure 10c. A good linear relationship was found, the correlation coefficient R^2^ being 0.9986. The calculated values are shown in Table 4. The results indicated that ΔG was negative at 298, 313, and 353 K, indicating that NH_3_ spontaneously adsorbed to the composite. The enthalpy change ΔH was −49.49 kJ·mol^−1^, indicating that the adsorption process was exothermic [79]. ΔS was <0 J·K^−1^, indicating that adsorption decreased the entropy. Adsorption of NH_3_ by the composite would decrease the degree of freedom, i.e., decrease the degree of disorder in the system. Besides, the equivalent heat of adsorption of ammonia on PA-PEI-MOF303(Al) was calculated with a magnitude of 30.44–62.77 kJ·mol^−1^ (Figure 10d), indicating that the adsorption occurs by physical adsorption (<40 kJ·mol^−1^) and chemisorption (40–300 kJ·mol^−1^) [80], consistent with our conclusions from the adsorption-desorption curves (Figure 10a). The isosteric heat of adsorption decreases logarithmically with increasing adsorption due to the inhomogeneity of the solid surface, which preferentially adsorbs on high-potential energy surfaces, releasing greater amounts of heat.

#### 3.4.3. Breakthrough Tests

NH_3_ dynamic adsorption experiments were performed by breakthrough tests and the reversibility of NH_3_ adsorption by PA-PEI-MOF303(Al) was evaluated. As shown in Figure 11a, the PA-PEI-MOF303(Al) composite had an excellent adsorption capacity for NH_3_, and NH_3_ started to break through at 620 min for the first adsorption. As the number of regenerations increases, the breakthrough time decreases. The breakthrough time decreased by approximately 10% after the composite had been regenerated twice and it nearly had no change from two to ten regenerations. This was due to the existence of chemisorption between a small amount of ammonia gas and the composite material during chemisorption. Under the regeneration conditions of 423 K in vacuum, there existed a small amount of ammonia gas based on chemisorption that was difficult to remove. Then, the XRD of PA-PEI-MOF-303(Al) composite after NH_3_ adsorption was taken to examine the stability of the composite (Figure 11b). The XRD indicated that no significant changes in the crystalline structure of the PA-PEI-MOF303(Al) composite occurred after various numbers of NH_3_ adsorption-regeneration cycles at the same relative humidity. This indicated that the PA-PEI-MOF303(Al) composite had good stability for regeneration and could be reused multiple times.

### 3.5. NH_3_ Adsorption Mechanism

The mechanism through which NH_3_ adsorbed to the PA-PEI-MOF(Al) composite was investigated by comparing NH_3_ adsorption to the MOF303(Al) and PA-PEI-MOF(Al). Spectra acquired during adsorption and heating-desorption were analyzed by two-dimensional correlation FTIR spectroscopy (2DCOS-FTIR) [81]. Synchronous correlation spectra were used to identify changes in the functional groups and the rates at which such changes occurred. Asynchronous correlation spectra were used to correlate this information with the sequence of events, i.e., the sequence of changes.

#### 3.5.1. NH_3_ Adsorption Stage

XPS before and after NH_3_ adsorption were acquired to provide evidence for multi-site hydrogen bonds forming between the PA-PEI-MOF(Al) composite and NH_3_. It can be seen from Figure 12a that the binding energy of the Al 2p orbital in the PA-PEI-MOF303(Al) composite was 74.41 eV before NH_3_ adsorption and 74.02 eV after NH_3_ adsorption. This indicated that NH_3_ adsorbed to the Al–O clusters in the PA-PEI-MOF303(Al) composite and affected the Al 2p binding energy [25]. The N 1s orbital binding energy in the PA-PEI-MOF(Al) composite could be divided into energies of 399.57 and 401.06 eV (Figure 12b), which were attributed to C–NH and C=N, respectively, on the pyrazole ring [67]. After NH_3_ adsorption, the binding energies for the C–NH and C=N sites had decreased to 399.07 and 400.62 eV, respectively, indicating that C–NH and C=N on the pyrazole ring readily formed hydrogen bonds with the captured NH_3_.

The synchronous 2DCOS-FTIR map of MOF303(Al) is shown in Figure 13a. Five obvious auto-peaks were found at 986, 1020, 1080, 1220, and 1370 cm^−1^. These were assigned to C–O, N–H, and NH_3_ absorption [65]. The cross-peaks at 1370 cm^−1^ (NH_3_) and 1220 cm^−1^ (partial N–H intra-planar wobbling vibrations) changed in opposite directions, indicating that strong hydrogen bonding occurred between NH_3_ and the N–H sites on the MOF, i.e., H–N(MOF)···H(NH_3_) bonds formed [82]. The 1370 cm^−1^ peak had marked negative cross-peaks at 986 and 1020 cm^−1^. The stronger signal indicated that the hydroxyl group (Al/C–O(MOF)···H(NH_3_)) made a strong contribution to NH_3_ adsorption [25,83]. These results and the asynchronous spectrum shown in Figure 13b indicated that the functional group peaks changed in the order 986 cm^−1^ > 1020 cm^−1^ > 1370 cm^−1^ > 1200 cm^−1^, indicating that the Al/C–O structure in the composite preferentially adsorbed NH_3_ throughout the reaction. After NH_3_ adsorption, NH_3_ formed intramolecular/intermolecular hydrogen bonds with N–H, which changed the conformation of the whole MOF and shifted the functional group peaks to varying degrees. The synchronous 2DCOS-FTIR map of PA-PEI-MOF303(Al) indicated that a new weak auto-peak appeared at 1630 cm^−1^ (Figure 13c), indicating that C=O in the composite contributed to NH_3_ adsorption and that these active sites were more abundant in PA-PEI-MOF303(Al) than in MOF303(Al). This and the asynchronous profile (Figure 13d) indicated that the activities of the adsorption sites decreased in the order Al/C–O > N–H > C=O and that the Al–O–Al structure was the most active adsorption site in the PA-PEI-MOF303(Al). C=O was found to make some contribution to NH_3_ adsorption but to be less active than the Al–O and N–H structures. In conclusion, multiple hydrogen bonding locations (N–H, Al/C–O, and C=O) between the MOFs and NH_3_ in both the MOF303(Al) and PA-PEI-MOF303(Al) were found to be key for efficient NH_3_ adsorption and desorption, and many more adsorption sites and active N–H sites were found in PA-PEI-MOF303(Al) than in MOF303(Al).

#### 3.5.2. NH_3_ Heating−Desorption Stage

The diagrams shown in Figure 14a,b indicated that the MOF303(Al) and PA-PEI-MOF303(Al) groups changed markedly during heating-desorption. For example, the peak for stretching vibrations of COO– in the MOF303(Al) at 1414 cm^−1^ and a characteristic peak shift for nitrogen heterocycles (e.g., the pyrazole ring) clearly changed. This would have been caused by the high temperature causing NH_3_ desorption and destroying COO–(Al–O) clusters and hydrogen bonds between N–N and NH_3_ [84]. The position shifts and intensity changes for the N–H and C–N bands at 1500–1600 cm^−1^ indicated that the structures mentioned above may also have been NH_3_ adsorption sites. The relative intensities of the peaks related to hydroxyl groups and hydrogen bonds (3424 and 1079 cm^−1^) in PA-PEI-MOF303(Al) were also strongly decreased by heating. The loss of NH_3_ from PA-PEI-MOF303(Al) caused marked changes in intermolecular hydrogen bonds, particularly the disappearance of the characteristic NH_3_ peaks at 1230 and 1354 cm^−1^, and this was strong proof for efficient NH_3_ loss [85]. However, this could be attributed to changes in interactions between the host and guest at high temperatures, but the spectral characteristics did not change markedly, indicating that the material did not decompose at 423 K.

Six distinct auto-peaks were found for MOF303(Al) during the heating-desorption phase in the synchronous 2DCOS-FTIR map of MOF303(Al) (Figure 15a). The peaks at 986, 1020, 1080, 1370, 1400, and 1540 cm^−1^ mainly represented Al/C–O, N–H, NH_3_, COO–, and other groups. The intensity of the NH_3_ peak at 1370 cm^−1^ decreased as the temperature increased, indicating strong NH_3_ desorption. As the intensity of the peak at 1370 cm^−1^ decreased, the intensities of the peaks at 986, 1020, and 1540 cm^−1^ increased. This was mainly attributed to NH_3_ loss leading to reversal of the energy band position and an increase in intensity caused by NH_3_ loss. The asynchronous pattern (Figure 15b) indicated that the hydrogen bonds between Al/C–O and NH_3_ (Al/C–O(MOF)···H(NH_3_)) were preferentially broken, indicating that the Al–O–Al and C–OH structures efficiently and reversibly adsorbed NH_3_ but were influenced by the lone pairs of electrons in N–H, and that the strong hydrogen bonds H–N(MOF)···H(NH_3_) were less active in terms of reversibility. The synchronous 2DCOS-FTIR map of PA-PEI-MOF303(Al) (Figure 15c) was essentially the same as that of MOF303(Al), but analysis in combination with the asynchronous mapping signal (Figure 15d) indicated that C=O(MOF)···H(NH_3_) hydrogen bonding was more stable and desorption occurred only at high temperatures.

## 4. Conclusions

High-density uniform growth of MOF303(Al) on PA-PEI substrate was achieved using a stepwise impregnation LBL growth method. This method promotes the nucleation growth of MOF 303(Al) in the pores of PA-PEI substrate. Characterization tests indicated that the PA-PEI-MOF303(Al) composite had a hierarchical pore favorable for adsorption and mass transfer. Adsorption tests indicated that the PA-PEI-MOF303(Al) composite had a high adsorption capacity for NH_3_ (16.07 mmol·g^−1^ at 100 kPa and 298 K), which was two-three times higher than the adsorption capacities of conventional activated carbon and other materials. Excellent reversibility was found in regeneration penetration cycle tests (approximately 80% protective time at 298 K and relative humidity <5%). The adsorption mechanism of the composite was investigated by in situ FTIR spectroscopy, which indicated that the Al–O–Al/C–OH, N–H, and –OH groups in the PA-PEI-MOF303(Al) formed multi-location hydrogen bonding interactions with NH_3_ and that Al/C–O(MOF)···H(NH_3_) were the main bonds involved in efficient NH_3_ adsorption. The composite had more active sites (e.g., C=O) than did MOF303(Al), which may have been an important reason the composite could efficiently adsorb NH_3_.

## Figures and Tables

**Figure 1 nanomaterials-13-00727-f001:**
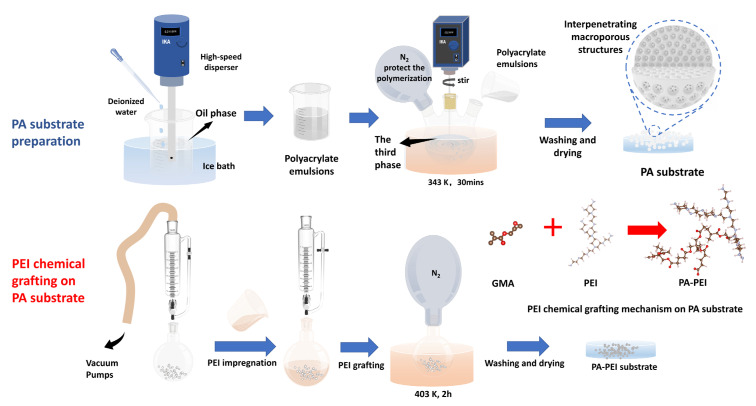
The reaction process of the synthesis of PA and PA-PEI.

**Figure 2 nanomaterials-13-00727-f002:**
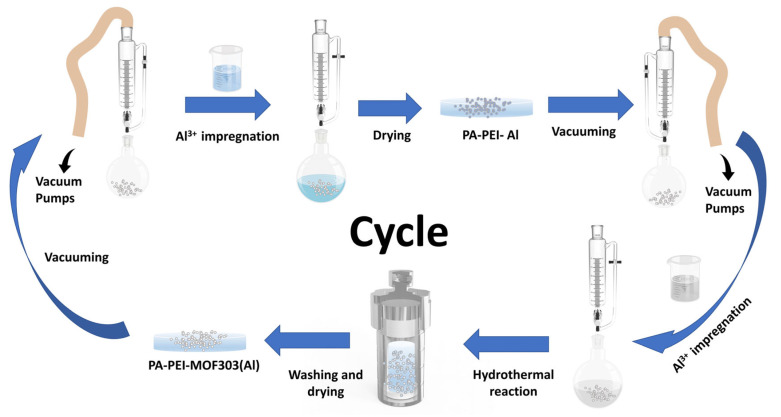
The preparation progress of PA-PEI-MOF303(Al) by stepwise impregnation LBL growth method.

**Figure 3 nanomaterials-13-00727-f003:**
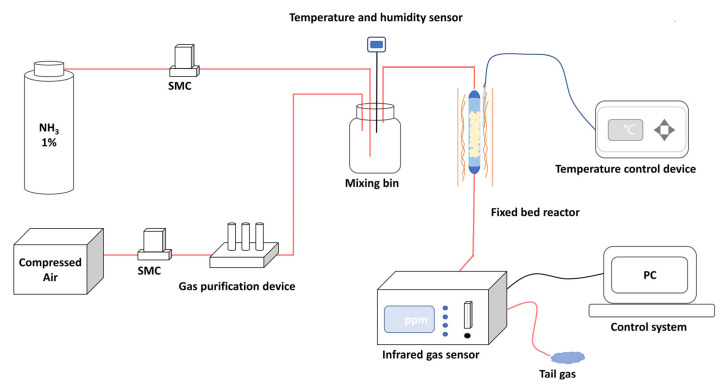
Schematic diagram of the NH_3_ dynamic adsorption evaluation system.

**Figure 4 nanomaterials-13-00727-f004:**
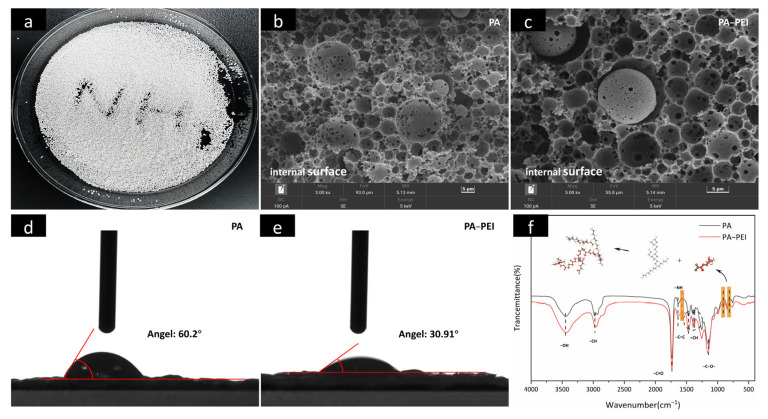
Optical picture of PA (**a**); SEM images of the cross-sectional view of PA (**b**) and PA-PEI (**c**); contact angle of PA (**d**) and PA-PEI (**e**); FTIR spectra of PA and PA-PEI (**f**).

**Figure 5 nanomaterials-13-00727-f005:**
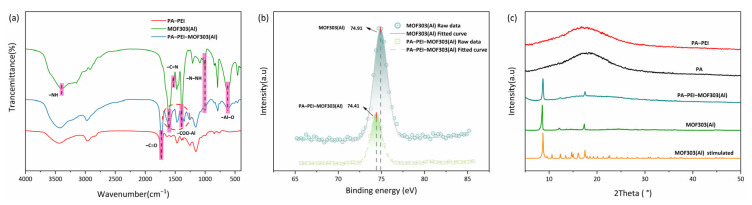
FTIR spectra (**a**), XPS spectra (**b**), and XRD patterns of PA, PA-PEI, MOF303(Al), and PA-PEI-MOF303(Al) (**c**).

**Figure 6 nanomaterials-13-00727-f006:**
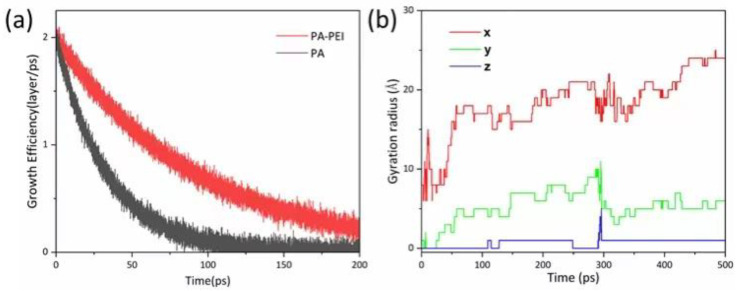
(**a**) Growth efficiency simulation of MOF303(Al) on PA and PA-PEI surface; (**b**) gyration radius of MOF303(Al) on PA-PEI substrates.

**Figure 7 nanomaterials-13-00727-f007:**
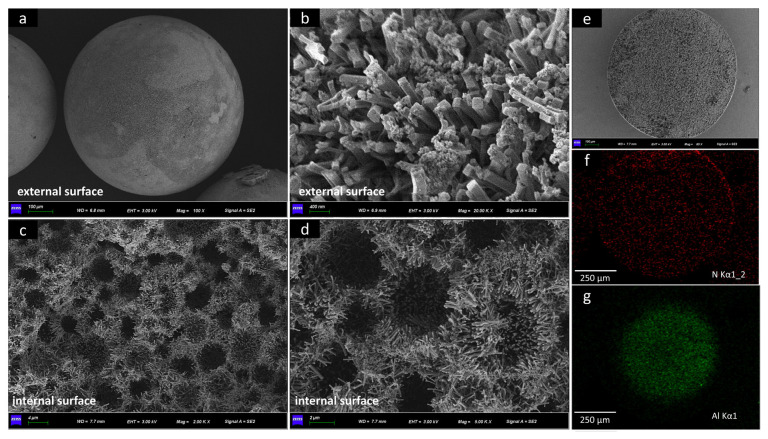
(**a**–**d**) SEM images of PA-PEI-MOF303(Al); (**e**–**g**) EDS maps of N and Al elements in PA-PEI substrate.

**Figure 8 nanomaterials-13-00727-f008:**
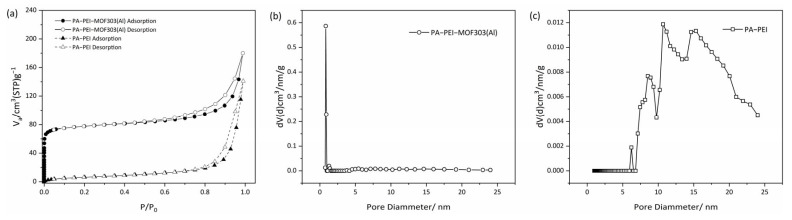
(**a**) N_2_ adsorption-desorption isotherms and (**b**,**c**) pore size distribution curves of (**b**) PA-PEI-MOF303(Al) and (**c**) PA-PEI.

**Figure 9 nanomaterials-13-00727-f009:**
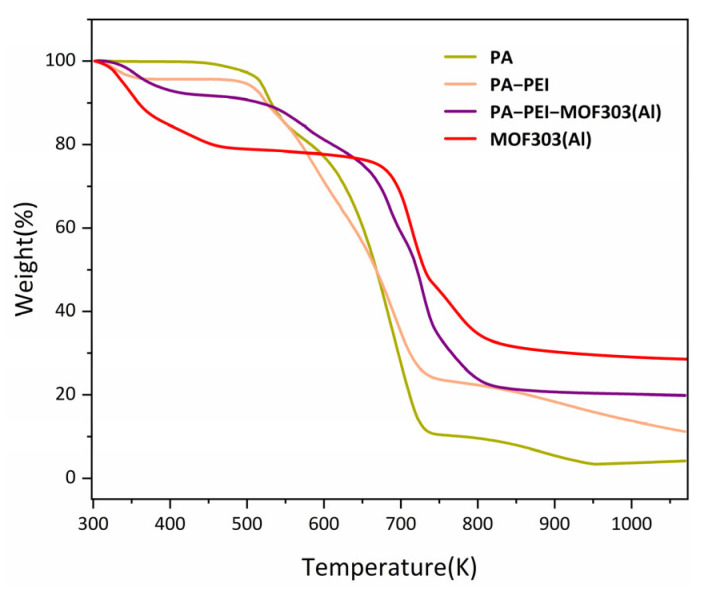
TGA curves of the prepared materials.

**Figure 10 nanomaterials-13-00727-f010:**
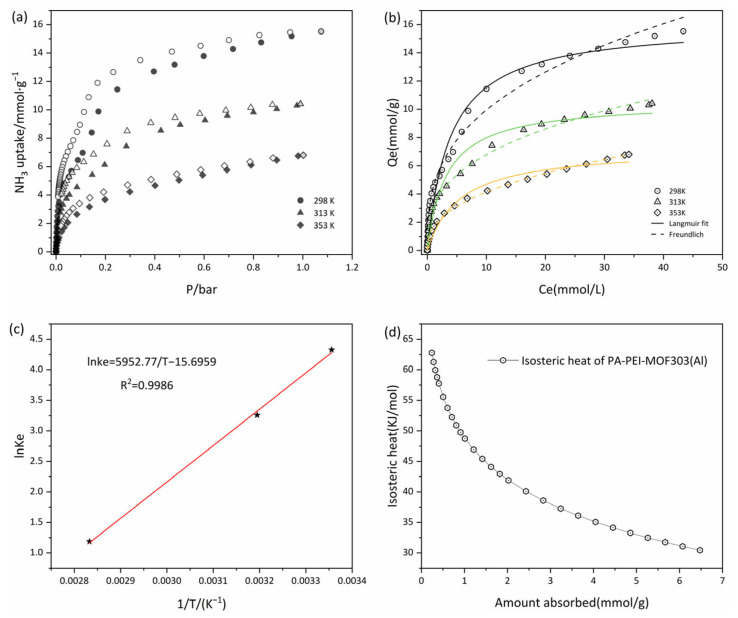
(**a**) Adsorption-desorption isotherms at 298, 313, and 353 K; (**b**) isothermal fitting of adsorption at 298, 313, and 353 K; (**c**) plots of ln Ke° versus 1/T for estimating thermodynamic parameters of PA-PEI-MOF(Al); (**d**) the isosteric heats of NH_3_ adsorption on PA-PEI-MOF(Al).

**Figure 11 nanomaterials-13-00727-f011:**
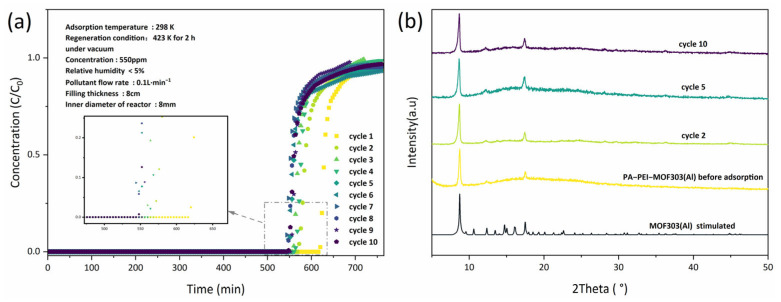
(**a**) NH_3_ breakthrough curves of PA-PEI-MOF(Al) for different adsorption-desorption cycle times; the inset is a magnified image of the tagged area; (**b**) XRD patterns of PA-PEI-MOF(Al) before and after adsorption.

**Figure 12 nanomaterials-13-00727-f012:**
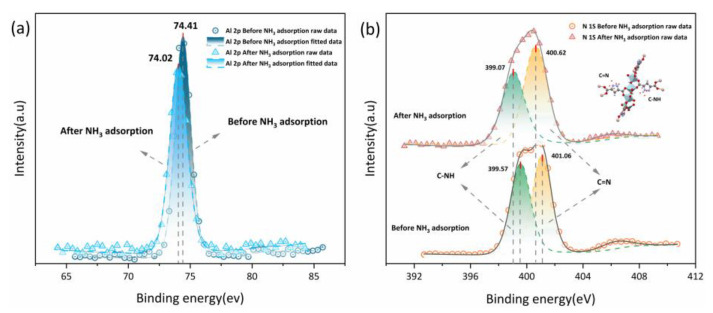
(**a**) Al 2p XPS spectra and (**b**) N 1s XPS spectra of PA-PEI-MOF303(Al) before and after NH_3_ adsorption.

**Figure 13 nanomaterials-13-00727-f013:**
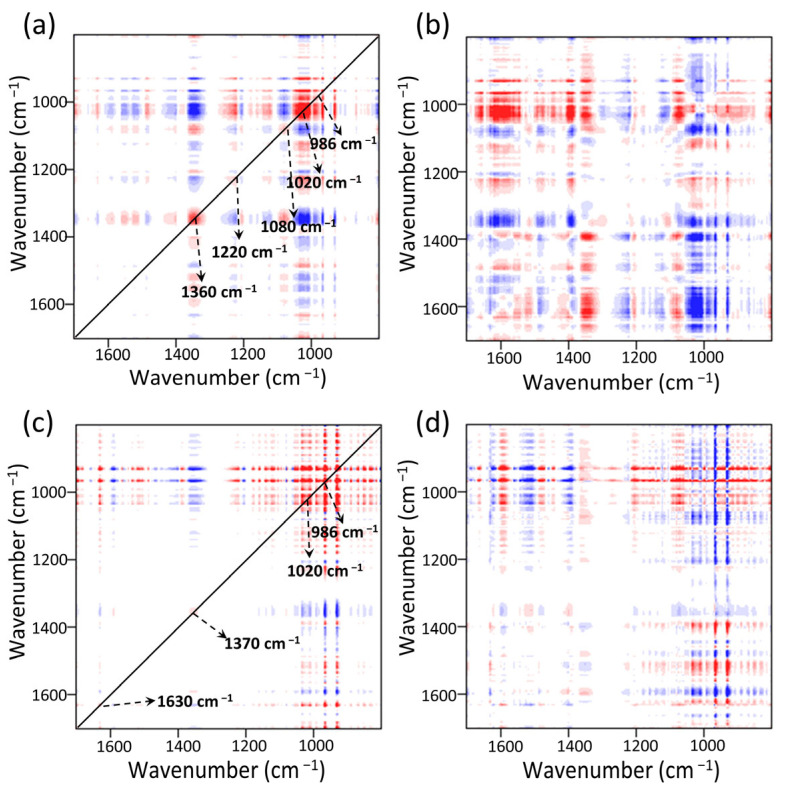
NH_3_ adsorption stage: (**a**,**b**) synchronous (**a**) and asynchronous (**b**) 2DCOS-FTIR maps of MOF303(Al); (**c**,**d**) synchronous (**c**) and asynchronous (**d**) 2DCOS-FTIR maps of PA-PEI-MOF303(Al). Red represents positive correlations and blue represents negative correlations; a higher color intensity indicates a stronger positive or negative correlation.

**Figure 14 nanomaterials-13-00727-f014:**
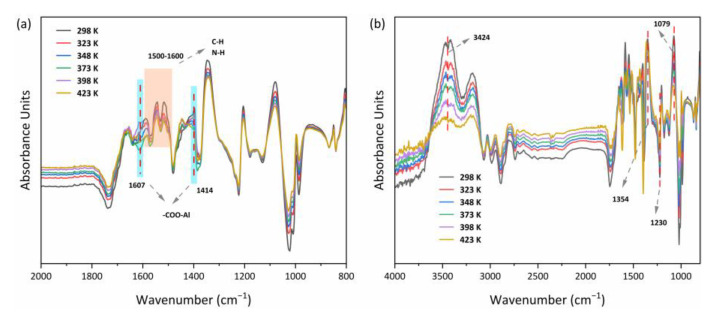
In situ FTIR spectra of MOF303(Al) (**a**) and PA-PEI-MOF303(Al) (**b**).

**Figure 15 nanomaterials-13-00727-f015:**
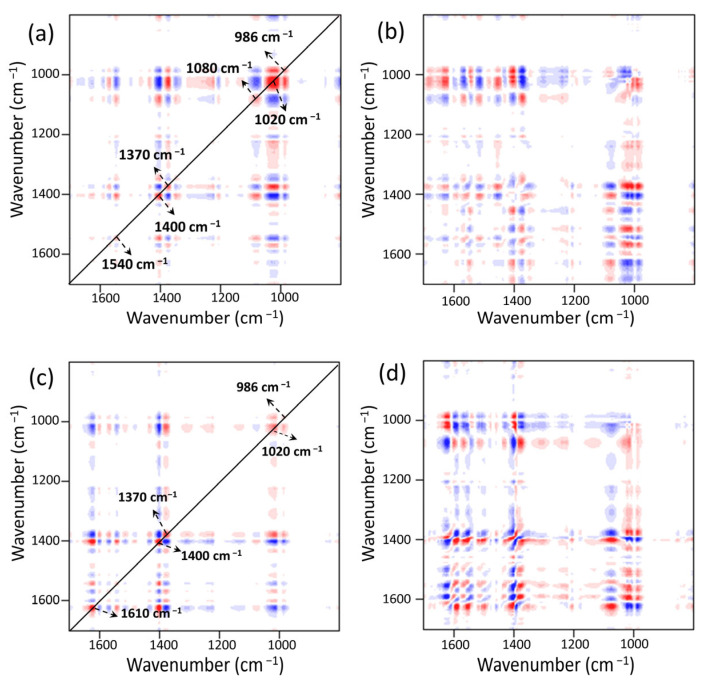
NH_3_ heating−desorption stage: (**a**,**b**) synchronous (**a**) and asynchronous (**b**) 2DCOS-FTIR maps of MOF303(Al); (**c**,**d**) synchronous (**c**) and asynchronous (**d**) 2DCOS-FTIR maps of PA-PEI-MOF303(Al). Red represents positive correlations and blue represents negative correlations; a higher color intensity indicates a stronger positive or negative correlation.

**Table 1 nanomaterials-13-00727-t001:** Physical parameters of PA-PEI and PA-PEI-MOF303(Al).

	BET Specific Surface Area	Micropore Volume	Total Pore Volume	Micropore Size Distribution
	S_BET_ [cm^2^·g^−1^]	V_m_ [cm^3^·g^−1^]	V_t_ [cm^3^·g^−1^]	D [nm]
PA-PEI	24.2	/	0.22	/
PA-PEI-MOF303(Al)	302.8	0.10	0.28	0.45–0.80

**Table 2 nanomaterials-13-00727-t002:** NH_3_ capture performances on various porous materials.

Material Type	Sample	NH_3_ Adsorption Capacity (mmol·g^−1^)	Regeneration Condition	Reference
MOF	M(NA)_2_ (M = Zn, Co, Cu)	6.00–17.50 (298 K)	423 K for 70 min under vacuum	[73]
MOF	M-2(INA) (M = Cu, Co, Ni, Cd)	12.00–13.00 (298 K)	423 K for 50 min	[74]
MOF	MOF303(Al)	19.70 (298 K)	423 K for 2 h under vacuum	[25]
COF	COF-10	15.00 (298 K)	473 K for 12 h under vacuum	[15]
Zeolite	Zeolite-A	8.39 (295 K)	Vacuum	[75]
Zeolite	Zeolite-13X	9.00–9.30 (room temperature)	Not mentioned	[76]
Activated carbon	Activated carbon	6.36 (298 K)	323 K for 5 h under vacuum	[77]
MOF composite	PA-PEI-MOF303(Al)	16.07 (298 K)	423 K for 2 h under vacuum	This work

**Table 3 nanomaterials-13-00727-t003:** PA-PEI-MOF303(Al) Langmuir and Freundlich fitting parameter.

Sample	Langmuir			Freundlich		
	$Q_{m}/\mathrm{mmol}\cdot g^{-1}$	$K_{L}/L\cdot {\mathrm{mmol}}^{-1}$	*R* ^2^	${k_{f}}_{\;}$	${n_{f}}_{\;}$	*R* ^2^
298 K	16.07	0.2558	0.9611	4.4901	0.3455	0.9889
313 K	10.53	0.3132	0.9637	3.0802	0.3430	0.9950
353 K	7.28	0.1815	0.9745	1.6296	0.4042	0.9984

**Table 4 nanomaterials-13-00727-t004:** Thermodynamics parameters for NH_3_ adsorption of PA-PEI-MOF303(Al).

Temperature (K)	$\Delta G\;(\mathrm{KJ}\cdot {\mathrm{mol}}^{-1})$	$\Delta S\;(J\cdot {\mathrm{mol}}^{-1}\cdot K^{-1})$	$\Delta H\;(\mathrm{KJ}\cdot {\mathrm{mol}}^{-1})$
298	−10.77	−130.50	−49.49
313	−8.54		
353	−3.55		

## Data Availability

Not applicable.

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
