# Peer review of "Fabrication of PA-PEI-MOF303(Al) by Stepwise Impregnation Layer-by-Layer Growth for Highly Efficient Removal of Ammonia"

_nanomaterials, 2023, doi:10.3390/nano13040727_

Round 1
Reviewer 1 Report
There are some font inconsistencies. For example, in the title of section 2.2.1 and a few other places. Some English editing is required.
If only the PA-PEI-MOF303(Al)-cycle-5 was selected to be tested, there is no need to introduce the other cycles and the in situ growth method in the main text. Those details should be kept in the supporting information section.
In section 3.3.2 the authors claim “No adsorption occurred in the low-pressure region, and an obvious hysteresis loop during the adsorption and desorption process was found, indicating that the PA-PEI contained mesopores.” It is pretty clear there is adsorption in the low-pressure region. And the mesopores need to be calculated using the BJH model to show the mesoporosity of this material.
Figure 9 claims to have curves of PA-PET and PA-PEI-MOF303(Al)-5-cycle, but it seems only one set of isotherms in the figure. Thus when the author claims “The BET specific surface area of the PA-PEI substrate was 24.2 m2·g-1, and no micropores were found” there are no data to refer to. Also, pore sizes smaller than 2 nm are all considered micropores.
From Figure 11, there’s enough data to calculate the heat of adsorption vs adsorbed amount figure. This would be very informative to understand the NH3 adsorption behavior.
The author claims The ΔH value was <-40 kJ·mol-1, indicating that adsorption occurred through strong physical adsorption and weak chemical adsorption. But the ΔH value calculated was -49 kJ·mol-1. This is a contradictory statement.
Can the author discuss the weight percentage of MOF303 in the composite material?
Reviewer 2 Report
This work is devoted to the synthesis and detailed investigation of a composite MMM membrane of polymer and a porous MOF. 303 (Al) for adsorption of ammonia.
while the idea of making membranes to overcome the troubles of the crystalline-powder like materials is not new, it is a rapidly developing and important field and the authors achieved excellent results by using a layer-by-layer method. The goal here was to find a a way increase the loading of the MOF within the matrix. Moreover, the resulting material had some additional sites. The adsorption process as well as the molecular mechanism was investigated by a set of physical tools and seems to show a consistent story.
So i think, the paper is qualified for being published after minor corrections:
1. - which is the seurface are and pore volumes for the pristine MOF? i haven't noted the number, only for the composites....
2. - in regard to the breakthrough curves - the stable situation is reach after the 2-3 cycle. Why there is a change between the 1 and 3 cycles? polymer degradation? desorption of residuals from the synthesis during the regeneration process?
the authors should at least make some assumptions on this!
3. - my additional concern is the 2D COS correlation spectroscopy.
First of all the authors must clarify that it is a 2DCOS-FTIR spectroscopy.
secondly, i think it will be much better and clearer for the audience of the journal if the authors introduce an additional paragraph where the detailed methodology how these correlation images were created would be introduced. The 2DCOS spectroscopy is not that frequent actually and especially chemist might have difficulties to understand what was done and how these 2D plot were generated.
I think this is quite important.
4. Also, i think, that it is important do add some more references on groups working on ammonia adsorption in porous MOFs - in particular Hiroshi Kitagawa from Kyoto university and others.
